# Intelligent Localization and Deep Human Activity Recognition through IoT Devices

**DOI:** 10.3390/s23177363

**Published:** 2023-08-23

**Authors:** Abdulwahab Alazeb, Usman Azmat, Naif Al Mudawi, Abdullah Alshahrani, Saud S. Alotaibi, Nouf Abdullah Almujally, Ahmad Jalal

**Affiliations:** 1Department of Computer Science, College of Computer Science and Information System, Najran University, Najran 55461, Saudi Arabia; afalazeb@nu.edu.sa (A.A.); naalmudawi@nu.edu.sa (N.A.M.); 2Department of Computer Science, Air University, E-9, Islamabad 44000, Pakistan; 200016@students.au.edu.pk (U.A.); ahmadjalal@mail.au.edu.pk (A.J.); 3Department of Computer Science and Artificial Intelligence, College of Computer Science and Engineering, University of Jeddah, Jeddah 23218, Saudi Arabia; asalshahrani2@uj.edu.sa; 4Information Systems Department, Umm Al-Qura University, Makkah 24382, Saudi Arabia; ssotabi@uqu.edu.sa; 5Department of Information Systems, College of Computer and Information Sciences, Princess Nourah Bint Abdulrahman University, Riyadh 11671, Saudi Arabia

**Keywords:** activity recognition, deep learning, deep neural decision forest, genetic algorithm, IoT, localization, recursive feature elimination, smartphone, smartwatch

## Abstract

Ubiquitous computing has been a green research area that has managed to attract and sustain the attention of researchers for some time now. As ubiquitous computing applications, human activity recognition and localization have also been popularly worked on. These applications are used in healthcare monitoring, behavior analysis, personal safety, and entertainment. A robust model has been proposed in this article that works over IoT data extracted from smartphone and smartwatch sensors to recognize the activities performed by the user and, in the meantime, classify the location at which the human performed that particular activity. The system starts by denoising the input signal using a second-order Butterworth filter and then uses a hamming window to divide the signal into small data chunks. Multiple stacked windows are generated using three windows per stack, which, in turn, prove helpful in producing more reliable features. The stacked data are then transferred to two parallel feature extraction blocks, i.e., human activity recognition and human localization. The respective features are extracted for both modules that reinforce the system’s accuracy. A recursive feature elimination is applied to the features of both categories independently to select the most informative ones among them. After the feature selection, a genetic algorithm is used to generate ten different generations of each feature vector for data augmentation purposes, which directly impacts the system’s performance. Finally, a deep neural decision forest is trained for classifying the activity and the subject’s location while working on both of these attributes in parallel. For the evaluation and testing of the proposed system, two openly accessible benchmark datasets, the ExtraSensory dataset and the Sussex-Huawei Locomotion dataset, were used. The system outperformed the available state-of-the-art systems by recognizing human activities with an accuracy of 88.25% and classifying the location with an accuracy of 90.63% over the ExtraSensory dataset, while, for the Sussex-Huawei Locomotion dataset, the respective results were 96.00% and 90.50% accurate.

## 1. Introduction

With the advancements in the field of artificial intelligence, many smart applications are being developed with every passing day that strive to embellish the lives of human beings. These applications cover the areas of entertainment, healthcare, indoor localization, smart homes, life-logging, rescue, and surveillance [1,2,3]. The availability of the internet is another plus that reinforces the development of such applications by providing the developers with access to tons of data. One such application is human activity recognition and localization, which sets its basis on the data acquired from the Internet of things (IoT) [4,5,6]. Modern smartphones and smartwatches contain various built-in sensors that provide information about the movements and positions of the user, and, if managed efficiently, the activity and location of the user can be accurately estimated. But processing the data from the built-in sensors of the aforementioned devices is no easy task, as the users are independent in how they use their smart devices [7]. They can hold their device, especially smartphones, in hand, or place them in a pocket or a bag. That enhances the sparsity of the data and makes the problem more challenging. Another challenge faced during the processing of IoT data is the noise in the signals. These sensors are highly sensitive to noise; sometimes, the data can be destroyed entirely and mislead the artificially intelligent model [8,9,10]. Overcoming these challenges requires innovative solutions integrating multiple sensor modalities, feature selection techniques, and advanced machine-learning algorithms. The successful development of accurate and reliable human activity recognition and localization systems has the potential to transform many aspects of our daily lives, making it an exciting and important research area [11].

For the problem under consideration, the sensor modalities that comprise the IoT system for this study include smartphones and smartwatches. These modalities provide the data of the accelerometer, gyroscope, magnetometer, global positioning system (GPS), and microphone. The accelerometer measures the translational force acting on the smartphone in the x, y, and z directions, which can be used to estimate the speed and direction of the movement [12]. The gyroscope senses the rotation of the smartphone about the x, y, and z directions, which proves beneficial in the estimation of the orientation. At the same time, the magnetometer gives information about the strength and direction of the magnetic field of the earth, that proves beneficial for the estimation of the absolute location of the user [13]. The microphone data also prove very helpful while guessing the location and activity of the user, as the microphone provides the sound data in the context of the activity being performed, like a person breathing heavily while running or the unique sound of the crowd while in a shopping mall. Last but not the least, GPS has proven its worth while localizing a person outdoors. But, when dealing with indoor spaces, GPS becomes vulnerable to noise, and its performance decays. Rather than disregarding the location data from GPS completely, it is reinforced by the aforementioned IoT sensors to estimate the user’s location with a good accuracy [14,15,16]. The use of more sensor modalities increases the diversity of the dataset so as to create a richer representation of the target classes. This proves to be a performance booster for the system and enhances the system’s ability to understand the context of an activity or location. Moreover, adding the sensors to the system having similar outputs can also prove useful in improving the fault robustness of the system, such that, when one sensor becomes unresponsive, the other sensor can take care of the event. With all these advantages, some limitations are also introduced in the system as the complexity of the system increases, and more time and power are consumed in the processing. Data processing challenges can also arise, causing the overall cost of the system to rise [17,18]. Therefore, it is essential to find a trade-off point when designing the system. Some of the inspirational works targeting the field of human activity recognition and localization are discussed below.

Numerous studies have been conducted in human activity recognition and localization using IoT data, each offering unique contributions and techniques. One framework proposed in [19] introduced a novel approach by employing a 1D-ResNet-SE model, which combines the standard 1D-ResNet architecture with a squeeze-and-excitation module. Through experimental evaluations, it was demonstrated that the 1D-ResNet-SE model outperformed other deep-learning techniques in terms of accuracy. This finding highlights the effectiveness of the proposed model in recognizing people’s actions in indoor localization scenarios. Another system, presented by Vesa et al. [20], focused on detecting complex activities by combining activity classification algorithms using a smartphone accelerometer and gyroscope data with a Beacon-based indoor localization predictor. The authors utilized a convolutional long short-term memory (ConvLSTM) algorithm for the multiclass classification of activities, while the positioning system involved an ensemble-based solution that integrated multilayer perceptron, gradient-boosted regression [21], and k-nearest neighbors. This comprehensive approach demonstrated promising results in accurately identifying human activities and facilitating precise localization. In the study conducted by Zhang et al. [22], a Wavelet-CNN deep-learning network was employed to pre-process and train data from the built-in sensors of smartphones, aiming to develop a robust human activity recognition (HAR) model. The researchers then utilized this model to identify different motion states of the target and adaptively adjusted pedestrian step detection and step size estimation algorithms. The proposed method exhibited considerable accuracy in human activity identification and localization. Yi et al. [23] proposed a smartphone-based indoor localization approach that integrated pedestrian dead reckoning with the Graph SLAM algorithm. They employed human activity classification to detect landmarks such as stairs and elevators during movement. The framework was designed to be invariant to the pose of the smartphone, ensuring robust observations regardless of how the user positioned the phone. The system improved activity inference based on temporal smoothness by utilizing a support vector machine for activity classification and incorporating a Bayesian framework with a hidden Markov model (HMM) [24]. Notably, the HMM jointly inferred the activity and floor information, enabling multi-floor indoor localization.

Xue et al. [25] designed a system to recognize human activities and fall detection. They used the concept of internal and external sensing. External sensing is based on external sensors like video data and smart home sensors [26], while, in internal sensing, smartphone sensors and wearable sensors are utilized. The study also provides an in-depth discussion about different types of hidden Markov models (HMMs) and their extensions. The extensions of HMM that are discussed in the study include the continuous density hidden Markov model (CDHMM) and hierarchical hidden Markov model (HHMM). A useful and adaptable approach for automatically segmenting and labelling biosignal data from wearable biosensors was put forth by Rodrigues et al. [27]. To view and assess the similarity of signal segments, it used a self-similarity matrix (SSM) based on feature representation. The system accurately segmented biosignals using change point detection and similarity measurements from the SSM, outperforming state-of-the-art techniques in terms of performance. It provided helpful insights for information retrieval and enabled cutting-edge applications like grouping and summarizing time-series profiles. It presented an intuitive and transparent approach that can be applied to single-channel or multimodal time-series data. A framework for using motion units (MUs) to simulate human actions was put forth by Liu et al. [28]. In the sensor-based activity detection system, MUs are the smallest, most easily identifiable units that constitute each human action. The study developed the Six-Directional Nomenclature (6DN) to give logical names to MUs. Five models were compared across three datasets, with an emphasis on activities of daily living (ADL) and sports-related activities, in order to assess the framework [29]. The study emphasized the value of performing repeated verification experiments and parameter adjustment while working on a model.

The proposed system focuses on human activity recognition and localization [30] using IoT data collected from smartphones and smartwatches. It employs a signal-denoising technique, i.e., a Butterworth filter, to enhance the input data’s quality. The system then divides the data into smaller windows using a hamming window, facilitating more sophisticated feature extraction. The windowed data are processed in parallel with feature extraction blocks for human activity recognition and human localization to extract robust features. A point worth noting is that the sensors are not equally beneficial for activity recognition and localization. Taking inspiration from the previous works, only the accelerometer, gyroscope, and magnetometer data are used for activity recognition, while, for localization, the global positioning system (GPS), microphone, and ambient sensor data are also added in the feature extraction process. After feature extraction, a recursive feature elimination algorithm is applied to independently select the most informative features from each category. The system performs the data augmentation using a genetic algorithm to enhance the robustness of the system to enhance its robustness and make it more generalizable. Finally, classifying activities and locations, a deep neural decision forest algorithm is used in parallel. The major limitations that the proposed system deals with include the orientation issues with the smartphone while the subject holds his smartphone or keeps it with him in a backpack or in his pocket, and the extraction of features that represent activities in a distinctive and efficient way so as to improve the performance of the system. The contributions of this study to the concerned research area are as follows:The system applies effective denoising techniques and windowing to improve the quality of sensor data collected from smartphones and smartwatches. It extracts meaningful features from the processed data, forming a solid foundation for accurate analysis.The system employs parallel feature extraction blocks dedicated to human activity recognition and human localization. This parallel processing approach captures relevant features simultaneously, enhancing the system’s accuracy and reliability in recognizing activities and locating the human subject.To improve feature selection, the system independently utilizes recursive feature elimination for activity recognition and localization modules. This iterative process selects the most informative features, reducing redundancy and noise. Additionally, a genetic-algorithm-based data augmentation technique generates diverse feature vectors, enhancing the system’s performance and generalization capabilities.The system uses advanced classification algorithms, such as the deep neural decision forest for activity classification and support vector machine for human localization. These algorithms provide powerful modeling capabilities, allowing the system to handle complex activity patterns and achieve a high accuracy in recognition and localization tasks.

The rest of this paper is organized as follows: Section 2 concisely assesses various cutting-edge approaches. The technique used by our proposed system is described in Section 3. Section 4 presents a quality assessment of our suggested technique on two benchmark datasets, a comparative analysis, and discussion. Finally, in Section 5, the study is wrapped up, and future directions are discussed.

## 2. Literature Review

Smartphone sensors have attracted significant attention among researchers in recent years. Various methods have been proposed to address the challenges of processing and analyzing sensor data to recognize human activities and localize them accurately. This section provides a deep insight into some of these methods.

### Human Activity Recognition and Localization Using IoT Data

Some of the previous works that have focused on recognizing human activities and human localization based on IoT data are shown in Table 1.

## 3. Materials and Methods

### 3.1. System Methodology

The proposed system aims to address human activity recognition and localization challenges by employing a robust and accurate framework. The system begins by pre-processing the input signal to enhance its quality. This is achieved by applying a second-order Butterworth filter, which effectively reduces noise and improves the overall signal fidelity. Following the denoising step, the pre-processed signal is windowed into smaller data chunks using hamming windows. The windowing process facilitates the extraction of meaningful features and enables more accurate analysis. Notably, multiple window stacks are generated, and each stack incorporates three windows. This approach enhances the reliability of the extracted features and contributes to the system’s overall performance. The stacked window data are then fed into two parallel feature extraction blocks, specifically designed for human activity recognition and human localization, respectively. For human activity recognition, only the inertial sensor data (i.e., accelerometer, gyroscope, and magnetometer) are utilized, and, for localization, additional data from microphone and GPS are also used. The use of microphone data enhances the system’s performance for indoor localization as the sound data prove to be an excellent support by providing data such as footstep sound, breathing sound, and other contextual sounds. By analyzing this data, rich contextual information is acquired that enhances the system’s performance manyfold. On the other hand, although GPS works best in outdoor environment, its performance decays in indoor environments. Yet, when the subject is near windows or in a semi-indoor environment, GPS data prove to be extremely useful for predicting the location. That is why these two sensors play a vital role in the architecture of the proposed system. Moving further with the architecture, in each feature extraction block, relevant features are extracted to capture the distinctive characteristics of the user’s activities and the human subject’s location. To further improve the system’s accuracy, a recursive feature elimination technique is applied independently of the features extracted from both modules. This technique aids in selecting the most informative features, effectively reducing redundancy and noise. After the feature selection process, a genetic algorithm generates ten generations of each feature vector. This approach serves as a form of data augmentation, enhancing the diversity and quality of the feature vectors. The augmentation process directly impacts the system’s overall performance, contributing to its effectiveness in activity recognition and human localization tasks.

A deep neural decision forest is implemented to classify human activity and location. This powerful classifier leverages the capabilities of deep neural networks [41] and decision trees to categorize the activities performed by the user accurately. The utilization of a deep neural decision forest enhances the system’s ability to handle complex and diverse activity patterns. By combining the components as mentioned earlier, the proposed system offers a comprehensive and sophisticated solution for human activity recognition and human localization, which is presented in Figure 1.

### 3.2. Pre-Processing

The proposed system takes one-dimensional signals from smartphone sensors as input, which are naturally vulnerable to the induction of noise from various sources such as electrical interference, mechanical vibration, or imperfect sensor calibration. This noise can obscure the underlying signal and make extracting meaningful information from the data difficult. The noise can also introduce errors and distortions in the signal, leading to inaccurate or unreliable results. Therefore, it is essential to pre-process and denoise the signal to improve the signal-to-noise ratio (SNR) before further analysis or processing. For this purpose, a second-order Butterworth filter [42] is used with a cutoff frequency of 0.001. In this way, the high-frequency components of the input signal that are usually caused by noise are rejected and the SNR of the input signal is improved. A comparison of raw and denoised signals is demonstrated in Figure 2.

### 3.3. Windowing

Windowing [43,44,45,46,47] of smartphone sensor data is a crucial step before the feature extraction phase. These techniques are used to partition a long signal into smaller, more manageable data chunks, allowing for more efficient analysis and extraction of relevant features. First, the signal is windowed using five-second intervals, and then three consecutive windows are stacked together to make a stacked segment of windows. This approach is useful in capturing the temporal information and, at the same time, improving the signal-to-noise ratio of the input signal. A hamming window [48,49] is used for the windowing process, which is defined as:(1)$$W_{n}=0.54-0.46Cos\left(\frac{2\pi n}{N-1}\right)\;$$
where $W_{n}$ is the value of the hamming window at index n, while N is the length of the window. A hamming window tapers the edges of the signal to zero, which helps to reduce spectral leakage when performing a Fourier transform. As a result, the spectrum of the signal has a smoother envelope and less distortion as compared to the case when using a rectangular window.

### 3.4. Feature Extraction—Human Activity Recognition

Multiple features are extracted for human activity recognition while utilizing the accelerometer, gyroscope, and magnetometer. The features include: maximum Lyapunov exponent (MLE), Mel-frequency cepstral coefficients (MFCC), embedding dimension (ED), fractal dimension (FD), skewness, and kurtosis. We extracted the hand-crafted features instead of using feature extraction libraries that automatically extract features so as to achieve a trade-off point between the computational complexity and the efficiency of the system [50,51,52]. All of these features are described in the context of human activity recognition below.

#### 3.4.1. Maximum Lyapunov Exponent

The maximum Lyapunov exponent (MLE) measures the rate at which two initially close trajectories in a nonlinear dynamical system diverge from each other. It provides information about the system’s sensitivity to small perturbations and can be used to characterize the chaotic behavior of the system. Mathematically, MLE is defined as the limit of the logarithm of the ratio of the distance between two nearby trajectories to their initial separation, as the separation goes to zero, averaged over all pairs of nearby trajectories [53,54]. More formally, given a dynamical system described by the set of differential equations:(2)$$f\left(x\right)=\frac{dx}{dt}$$
where x is a vector representing the state of the system, MLE is defined as:(3)$$\gamma =\underset{t\to \infty }{\lim}\frac{1}{t}ln(\frac{\left||\delta x(t)|\right|}{\left||\delta x(0)|\right|})$$
where γ is the MLE and δx(t) is the separation between two initially close trajectories at time *t*. The MLE is a handy feature for studying the behavior of nonlinear systems. HAR provides a measure of the degree of chaos or unpredictability in the movement patterns of the human body during different activities. The MLE extracted for different human activities is shown in Figure 3.

#### 3.4.2. Mel-Frequency Cepstral Coefficients

Mel-frequency cepstral coefficients (MFCCs) are widely used in human activity recognition due to their effectiveness in capturing the spectral characteristics of signals. The MFCCs are a set of features that represent the spectral envelope of a signal, based on the Mel scale. The Mel scale is a non-linear frequency scale that models the human auditory system’s perception of sound. The MFCCs are computed by taking the logarithm of the magnitude spectrum of the signal, followed by the application of the Mel filterbank, and then taking the discrete cosine transform (DCT) of the log filterbank energies. The resulting coefficients represent the signal’s spectral envelop in a compact and discriminative manner. Due to their ability to capture the relevant spectral information, MFCCs have been widely used in human activity recognition [55,56]. The MFCC plots for different activities are illustrated in Figure 4.

**Figure 3 sensors-23-07363-f003:**
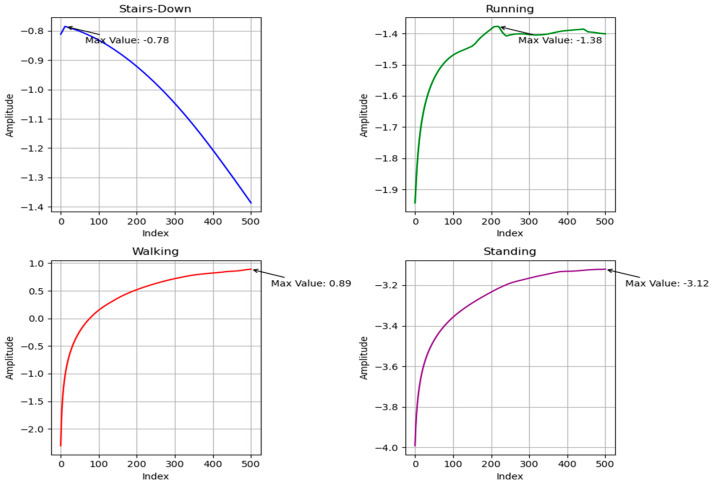
Maximum Lyapunov exponent for various activities of the ExtraSensory dataset.

**Figure 4 sensors-23-07363-f004:**
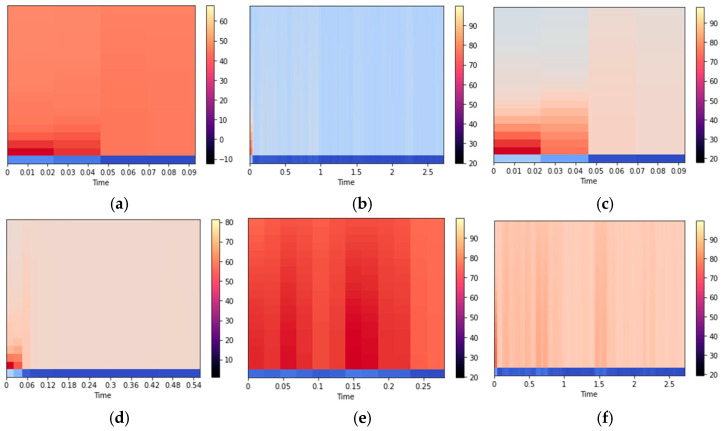
MFCC plot for (**a**) strolling, (**b**) sitting, and (**c**) bicycling from the ExtraSensory dataset, and (**d**) running, (**e**) walking, and (**f**) standing from the Sussex-Huawei Locomotion dataset.

#### 3.4.3. Fractal Dimension

Fractal dimension (FD) measures the complexity or roughness of a pattern that exhibits self-similarity at different scales. It is beneficial for characterizing the complexity of natural and artificial patterns and understanding the dynamics of physical and biological systems [57]. Intuitively, FD represents the rate at which the number of copies of the pattern increases as they are zoomed in. In the context of HAR, fractal dimension is used as a feature to represent the complexity of human motion patterns that gives the degree of self-similarity and nonlinearity in the movement patterns, which, in turn, provides precious information for the recognition of the performed activity. The fractal dimension of the input signal is defined as the exponent D in the following scaling relationship:(4)$$N\sim L^{D}$$
where N is the number of smaller copies of the signal needed to cover it at a given scale and L is the ratio of the size of the larger copy to the size of the more miniature copy. A representation of FD for various activities is given in Figure 5.

#### 3.4.4. Embedding Dimension

The embedding dimension (ED) holds significance within the realm of nonlinear dynamics and chaos theory, particularly in the context of time-series analysis. By constructing the phase space from time-series data, insights into the inherent dynamics of a given system can be analyzed. If the embedding dimension is shallow, essential information may be lost, yielding an inaccurate portrayal of the system’s behavior. Conversely, an excessively high embedding dimension may introduce spurious correlations and needlessly escalate computational complexity [58,59]. Various methodologies, such as the false nearest neighbor’s algorithm and the Cao method, reinforce estimating the embedding dimension. The false nearest neighbors’ algorithm has been used to find the embedding dimension in this case, due to its higher compatibility with our data. The embedding dimension plots for different activities are shown in Figure 6.

### 3.5. Feature Extraction—Human Localization

The features that were extracted for human localization include step detection, dynamic step-length estimation, heading direction estimation, and MFCCs using audio data. All of the details about these features are given below.

#### 3.5.1. Step Detection

This module detects the steps taken by the human subject at a particular location while utilizing the tri-axial accelerometer data [60]. The algorithm starts by calculating the magnitude of the acceleration while utilizing the equation given below:(5)$$magnitude=\sqrt{{acc\_x}^{2}+{acc\_y}^{2}+{acc\_z}^{2}}$$

While dealing with accelerometer, there is a gravity factor that needs to be rejected to obtain a reliable measurement. For that, the mean of the magnitude is subtracted from every point to obtain the net magnitude. In the next step, the mean of the net magnitude is calculated and is used as a threshold. Whichever instance of the net magnitude surpasses the threshold forms a peak in the waveform, and is considered a step. The steps detected for indoor and outdoor activities over the two datasets used in this study are shown in Figure 7.

#### 3.5.2. Dynamic Step-Length Estimation

Detecting the length of the steps taken by the human subject is another very important feature utilized in this study for human localization. The length of the steps taken by the human subject is estimated dynamically as the length of every step can be different from the previous one. To find the step length dynamically, left and right valley points are spotted around a particular peak point and the difference between the tagged timestamps of the valley points is calculated, which is regarded as the step length of that particular step [61]. Step-length estimation is depicted in Figure 8.

**Figure 7 sensors-23-07363-f007:**
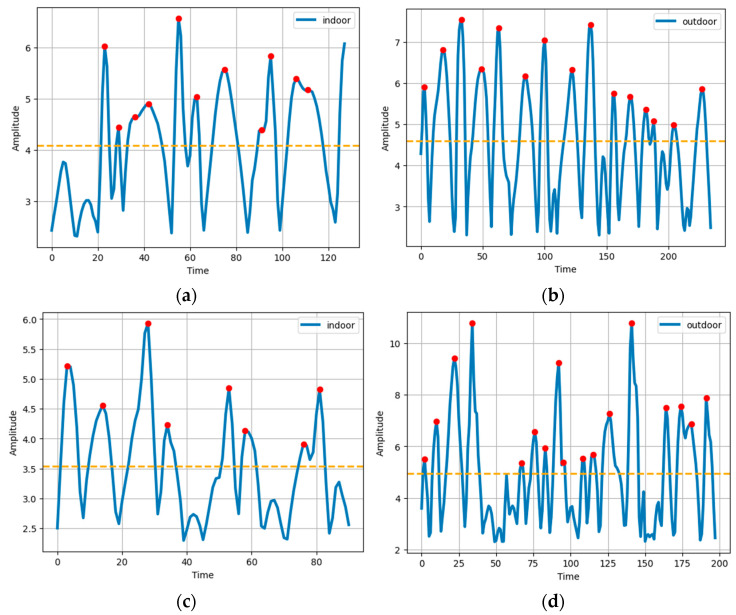
Steps for (**a**) indoor and (**b**) outdoor for the ExtraSensory dataset, and (**c**) indoor and (**d**) outdoor for the Sussex-Huawei Locomotion dataset.

**Figure 8 sensors-23-07363-f008:**
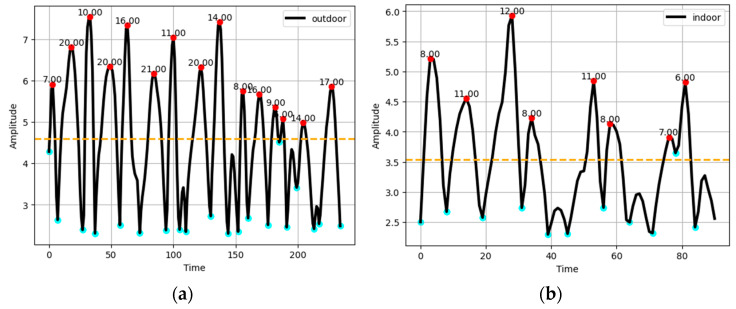
Step length for (**a**) outdoor for the ExtraSensory dataset, and (**b**) indoor for the Sussex-Huawei Locomotion dataset.

#### 3.5.3. Heading Direction Estimation

The heading direction of the subject is estimated using a combination of accelerometer, gyroscope, and magnetometer data [62,63]. To enhance the precision of the heading estimation, the estimations concerning the magnetometer and gyroscope are conducted independently, and then mean values of both of these heading estimates is considered as the final heading angle. First, the Euler angles including roll, pitch, and yaw are calculated using the triaxial accelerometer and magnetometer data. The governing equations for the calculation of roll (φ), pitch (θ), and yaw (Ψ) are given below:(6)$$\;\varphi ={tan}^{-1}(\frac{{Acc}_{y}}{{Acc}_{z}})$$
(7)$$\theta ={tan}^{-1}(\frac{-{Acc}_{x}}{\sqrt{{{Acc}_{y}}^{2}+{{Acc}_{z}}^{2}}})$$
(8)$$\;\Psi ={tan}^{-1}(\frac{M_{y}}{M_{x}})$$
where
(9)$$M_{y}={Mag}_{x}Sin\left(\varphi \right)Sin\left(\theta \right)+{Mag}_{y}\mathrm{Cos}\left(\varphi \right)-{Mag}_{z}Sin(\varphi )Cos(\theta )$$
(10)$$M_{x}={Mag}_{x}Cos\left(\theta \right)+{Mag}_{z}Sin(\theta )$$

For the calculation of the magnetometer heading $H_{M}$ at a specific instant of time, the rotation matrix and the magnetometer reading at that particular time instant undergo a matrix multiplication. The following equation represents the relation:(11)$$H_{M}=\left[\begin{array}{ccc} Cos(\varphi ) & Sin(\varphi )Sin(\theta ) & -Sin(\varphi )Cos(\theta )\\ 0 & Cos(\theta ) & Sin(\theta )\\ Sin(\varphi ) & -Sin(\theta )Cos(\varphi ) & Cos(\varphi )Cos(\theta ) \end{array}\right]\left[\begin{array}{c} {Mag}_{x}\\ {Mag}_{y}\\ {Mag}_{z} \end{array}\right]$$

Next, the gyro-quaternion for the estimation of the gyroscope-based heading $H_{G}$ is calculated using the mathematical equations given below:(12)$$\varphi^{*}={Gyr}_{x}+{Gyr}_{y}Sin\left(\varphi \right)\tan\left(\theta \right)+{Gyr}_{z}Cos(\varphi )tan(\theta )$$
(13)$$\theta^{*}={Gyr}_{y}Cos\left(\varphi \right)-{Gyr}_{z}Sin(\varphi )$$
(14)$$\Psi^{*}={Gyr}_{y}\left(\frac{Sin\left(\varphi \right)}{Cos\left(\theta \right)}\right)+{Gyr}_{z}(\frac{Cos(\varphi )}{Cos(\theta )})$$

Using the new angles, the gyro-quaternion matrix $G_{Q}$ is given by:(15)$$G_{Q}=\left[\begin{array}{c} Cos\left(\frac{\varphi^{*}}{2}\right)Cos\left(\frac{\theta^{*}}{2}\right)Cos\left(\frac{\Psi^{*}}{2}\right)+Sin(\frac{\varphi^{*}}{2})Sin(\frac{\theta^{*}}{2})Sin(\frac{\Psi^{*}}{2})\\ \begin{array}{c} Sin\left(\frac{\varphi^{*}}{2}\right)Cos\left(\frac{\theta^{*}}{2}\right)Cos\left(\frac{\Psi^{*}}{2}\right)-Cos(\frac{\varphi^{*}}{2})Sin(\frac{\theta^{*}}{2})Sin(\frac{\Psi^{*}}{2})\\ Cos\left(\frac{\varphi^{*}}{2}\right)Sin\left(\frac{\theta^{*}}{2}\right)Cos\left(\frac{\Psi^{*}}{2}\right)+Sin(\frac{\varphi^{*}}{2})Cos(\frac{\theta^{*}}{2})Sin(\frac{\Psi^{*}}{2})\\ Cos\left(\frac{\varphi^{*}}{2}\right)Cos\left(\frac{\theta^{*}}{2}\right)Sin\left(\frac{\Psi^{*}}{2}\right)-Sin(\frac{\varphi^{*}}{2})Sin(\frac{\theta^{*}}{2})Sin(\frac{\Psi^{*}}{2}) \end{array} \end{array}\right]$$

By using the $G_{Q}$ matrix, the rotation matrix R is calculated, which, finally, decides the heading direction of the subject. Rotation matrix is computed through the following mathematical procedure:(16)$$R=\frac{1}{2}\left[\begin{array}{c} \begin{array}{ccc} 0 & {-Gyr}_{x} & \begin{array}{cc} {-Gyr}_{Y} & {-Gyr}_{z} \end{array} \end{array}\\ \begin{array}{c} \begin{array}{ccc} {Gyr}_{x} & 0 & \begin{array}{cc} {Gyr}_{z} & {-Gyr}_{y} \end{array} \end{array}\\ \begin{array}{ccc} {Gyr}_{x} & {-Gyr}_{z} & \begin{array}{cc} 0 & {Gyr}_{y} \end{array} \end{array}\\ \begin{array}{ccc} {Gyr}_{z} & {Gyr}_{x} & \begin{array}{cc} {-Gyr}_{y} & 0 \end{array} \end{array} \end{array} \end{array}\right]\cdot G_{Q}$$

Finally, gyroscope-based heading is formulated by the following equation:(17)$$H_{G}=\frac{G_{x}}{G_{y}}$$
where
(18)$$G_{x}=2(R_{m0}R_{m3}+R_{m1}R_{m2})$$
(19)$$G_{y}=1-2({R_{m2}}^{2}+{R_{m3}}^{2})$$

By taking the mean of the corresponding values of $H_{M}$ and $H_{G}$, the final heading direction of the human subject is specified. The heading direction curves for different subject locations are shown in Figure 9 and Figure 10.

#### 3.5.4. Mel-Frequency Cepstral Coefficients for Localization

Mel-frequency cepstral coefficients (MFCCs) are already introduced in detail in the previous section. For localization, the MFCCs are calculated using the audio data [64]. The results for different locations given by both datasets used in this study are given in Figure 11.

### 3.6. Recursive Feature Elimination

Recursive feature elimination (RFE) is a feature selection technique used to identify the most relevant features for a problem under consideration [65,66]. The algorithm recursively removes features from the input data and fits a model on the remaining features until the optimal subset of features is obtained. An architectural diagram of RFE is given in Figure 12.

**Figure 11 sensors-23-07363-f011:**
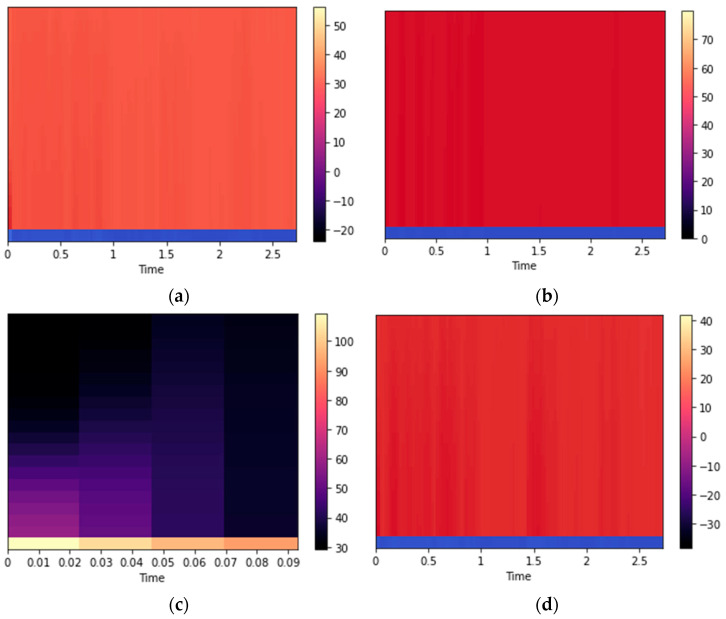
MFCCs for (**a**) at-home and (**b**) in-class locations from the ExtraSensory dataset, and (**c**) in-train and (**d**) in-car locations from the Sussex-Huawei Locomotion dataset.

**Figure 12 sensors-23-07363-f012:**
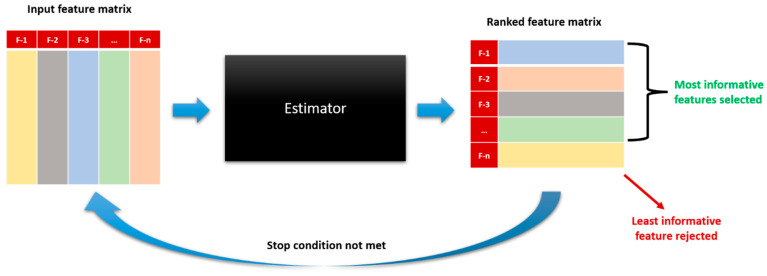
Block diagram for recursive feature elimination.

### 3.7. Data Augmentation

In data augmentation, a genetic algorithm (GA) is used to generate new data samples by randomly combining existing samples to create new ones [67,68]. The algorithm works such that it picks the first feature vector in the list and then randomly picks another feature vector from the remaining ones. Both vectors are split into three equal parts. The central parts of both of the vectors are then swapped. This resultant hybridized vector is regarded as the new generation of the first vector in the list. Then, the second feature vector is picked from the feature vectors’ list, and another random feature vector is selected from the list excluding the already-picked one. Following the described swapping process, the next generation is generated. The same process is carried out for all of the feature vectors one by one, and a total of ten generations are generated for each feature vector. All these data are then used for the training of the classification model. The graphical representation of GA as a data augmenter is given in Figure 13.

### 3.8. Classification

A deep neural decision forest (DNDF) is a powerful classification framework that comprises a deep neural network (DNN) [69] and a random forest [70], which is called decision forest (DF) in this case. It proved its excellence in the classification of human activities and locations where the input data represented complex patterns in high-dimensional data. It works in such a way that the deep neural network captures the high-level feature representations from the input and forwards it to the decision forest unit. Decision forest learns the pattern represented by the features and specifies the decision boundaries for the classification.

In the context of the proposed framework, DNDF is used in parallel to the classification of the human activities and locations. First, the respective features for both categories are fed to a DNN that takes a 740-dimensional feature vector as input and generates a 128-dimensional vector at its output. This 128-dimensional feature vector is then fed to a DF that learns the decision boundaries throughout the training process. A block diagram of a DNDF is demonstrated in Figure 14.

The DNDF classification model was trained on two datasets, i.e., the ExtraSensory dataset and SHL dataset, for human activity recognition and localization in parallel. Several hours were consumed in the training process of the model, while the memory used in training process was under 10 GB for all cases. The training time for activity recognition over the ExtraSensory dataset was 20,303 s (approximately 5.6 h), while the memory occupied by the weights was 9394.58 MB (approximately 9.4 GB), as shown by Figure 15. A similar plot is also shown in Figure 16, that shows that the memory usage for localization over the ExtraSensory dataset was 9443.99 MB, while the time the system took for training was 20,987 s.

Similar plots are shown in Figure 17 and Figure 18 regarding the time and memory consumption during the training of the model for activity recognition and localization over SHL dataset. Figure 17 shows that the runtime was 15,732 s, while the 6117.01 MBs of memory were utilized for activity recognition, while, according to Figure 18, 16,102 s were consumed to train for localization while utilizing 6825.23 MBs of computational space.

**Figure 14 sensors-23-07363-f014:**
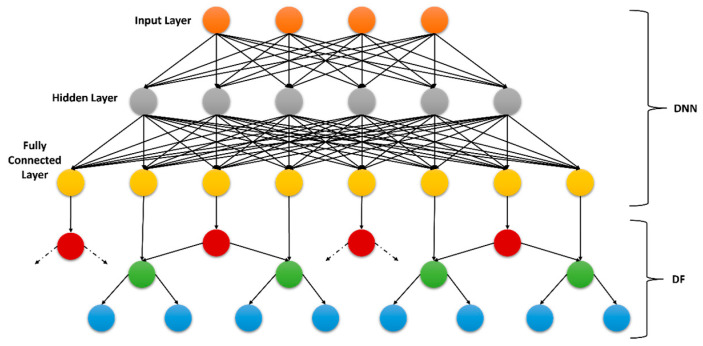
Block diagram for a deep neural decision forest classifier.

**Figure 15 sensors-23-07363-f015:**
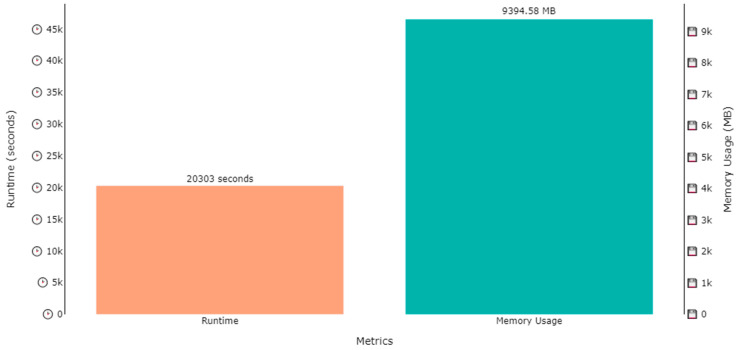
Runtime and memory usage plot for the training of a DNDF for activity recognition over the ExtraSensory dataset.

**Figure 16 sensors-23-07363-f016:**
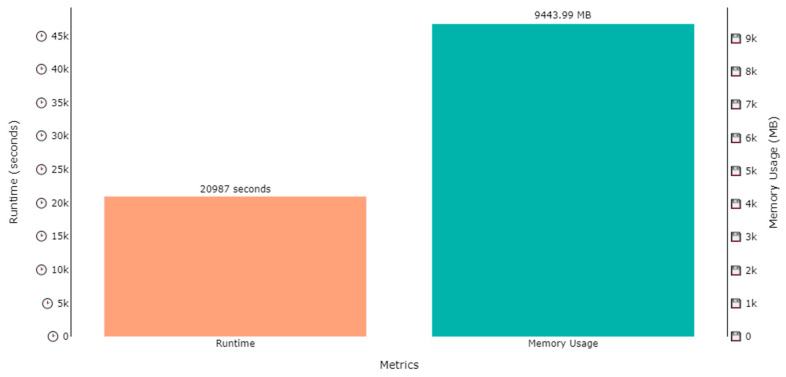
Runtime and memory usage plot for the training of a DNDF for localization over the ExtraSensory dataset.

**Figure 17 sensors-23-07363-f017:**
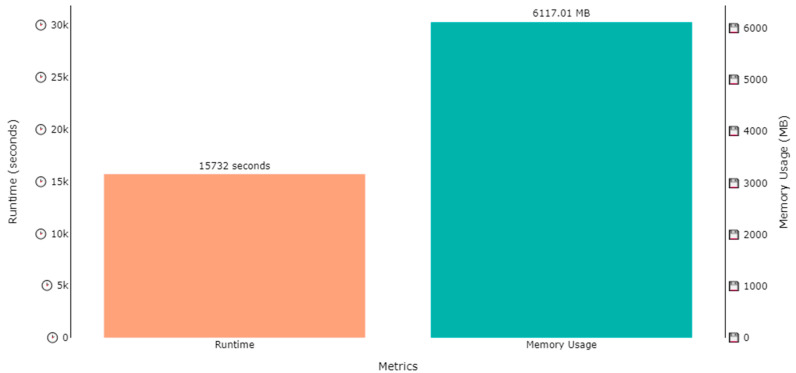
Runtime and memory usage plot for the training of a DNDF for activity recognition over SHL dataset.

**Figure 18 sensors-23-07363-f018:**
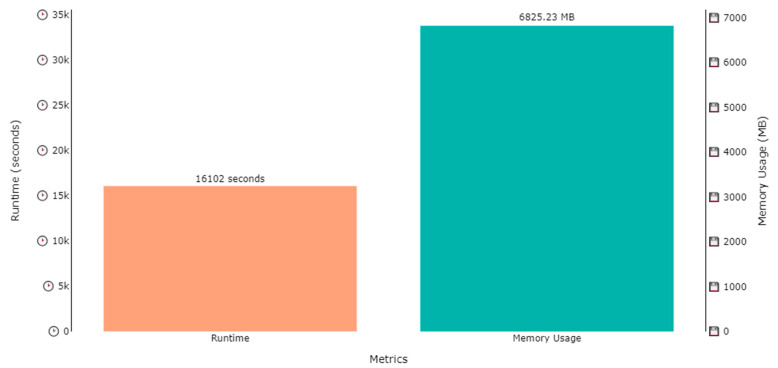
Runtime and memory usage plot for the training of a DNDF for localization over SHL dataset.

## 4. Experimental Setup and Evaluation

A laptop with an Intel^®^ CoreTM i7-7500U CPU @ 2.70 GHz and a 2.90 GHz processor, 16.0 GB of RAM, a 64-bit version of Windows 10, and Visual Studio Code as the coding environment was used to conduct this study. Two datasets were used in this study, i.e., the ExtraSensory and SHL datasets. Both of these datasets cover a diverse range of activities and locations. The reliability of the results was ensured by the 10-fold cross-validation process. We have made the proposed system’s code publicly available at GitHub, https://github.com/usman200016/IL-DHAR (accessed on 30 July 2023), for research purposes.

### 4.1. Dataset Descriptions

#### 4.1.1. ExtraSensory Dataset

The ExtraSensory dataset is a comprehensive collection of sensor data obtained from 60 users, each with a unique identifier (UUID), over a series of intervals, typically one-minute-long. The dataset is composed of measurements from sensors on users’ personal smartphones and smartwatches provided by the research team. The sensors used in the dataset include an accelerometer, gyroscope, magnetometer, watch accelerometer, location services, audio, and a watch compass. The human activities related to the dataset include sitting, lying down, standing, bicycling, running, strolling, stairs-up, and stairs-down, while the locations chosen for this study include indoor, at home, at school, at the workplace, outdoor, in class, at the gym, and at the beach [71].

#### 4.1.2. Sussex-Huawei Locomotion Dataset

The Sussex-Huawei Locomotion (SHL) dataset is a publicly available dataset that is designed to aid the development and evaluation of human activity recognition and localization algorithms. The dataset contains data from various sensors, including the accelerometer, gyroscope, magnetometer, and GPS. The dataset consists of 59 h of labelled data from four smartphones with diverse placement locations, including pocket, hand, torso, and bag. The human activities given in the dataset include sitting, walking, standing, and running, while the location includes indoor, outdoor, bus, train, subway, and car [72].

### 4.2. Experimental Results

#### 4.2.1. Experiment 1: Different Window Sizes on the ExtraSensory Dataset

We have performed additional experimentation with window sizes equal to 1, 2, 3, 4, 5, and 6 s, and we plot the pair plots of the features so as to analyze their linear separability. Figure 19 shows the pair plots of four features including the maximum Lyapunov exponent, MFCC coefficients, fractal dimension, and embedding dimension for the window sizes 1, 2, 3, 4, 5, and 6 s. The diagonal of each pair plot represents the distribution of four features, while the scatter plots of the examples of all eight activity classes related to the ExtraSensory dataset are shown at the off-diagonal positions. The overlap of the feature distributions shows their independency. The larger the overlap, the more the features depend upon each other, and vice versa. It can be examined from the plot that the most distinctive shapes of the feature distribution are produced while using a window size equal to 5 s. A possible reason for this can be the nature of the features and data that we are using such that the system responds to the five-second window in the best way.

#### 4.2.2. Experiment 2: Using the ExtraSensory Dataset

The proposed system was comprehensively evaluated by utilizing eight human activities and eight locations provided by the ExtraSensory dataset. A confusion matrix for the performance of the model for activity recognition over the ExtraSensory dataset is given in Table 2, which shows that “Lying Down” was the activity that was recognized with the highest confidence, i.e., 91%, while the model was at its lowest confidence, i.e., 86%, for the prediction of “Bicycling”. The overall score for the activity classification using a DNDF classifier happened to be 88.25%. Table 3 shows the individual precision, recall, and F1-score for all of the activities. A similar pattern like the confusion matrix can also be observed here, while the mean precision, mean recall, and mean F1-score was 89%, 89%, and 88%, respectively.

A confusion matrix was also plotted for the location classification, whose statistics are shown in Table 4. The highest recognition accuracy achieved by a location was 92% and it was shared among “Indoor”, “At school”, and “At beach”, while the lowest recognition accuracy was 89% and it was shared between the “Outdoor” and “At gym” locations. The mean accuracy of the localization was 90.63%. According to the classification report given in Table 5, the localization was 91% precise, while the recall and F1-score for the prediction of the location were 90% each.

**Figure 19 sensors-23-07363-f019:**
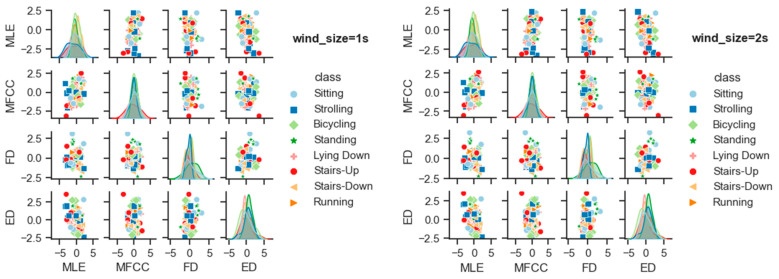
Comparison of the effect of window size on the linear separability of the features.

#### 4.2.3. Experiment 3: Using SHL Dataset

The SHL dataset consists of four human activities at six different locations. After training the model over the SHL dataset, it was comprehensively tested on it and the results were validated. Table 6 shows the confusion matrix for the model’s performance while recognizing the activities over the SHL dataset. “Running” was the activity that was recognized with the highest accuracy, i.e., 98%, while the lowest accuracy of the prediction was 94% for “Standing”. The system was 96% accurate on average while recognizing the activities belonging to the SHL dataset. Table 7 offers insights into the classification report of the system for activity classification. The mean precision and recall of the system were 96% each, while the F1-score was 95% on average.

The localization over the SHL dataset also produced promising results, which are shown by the confusion matrix in Table 8. Among a total of six different locations, “In car” was the location that was predicted correctly 96% of the time, which was the highest prediction accuracy. The lowest prediction accuracy was for “In Train”, i.e., 86%. The system showed a reliable performance by predicting every class with 90.50% correctness on average. For the model’s classification report which is given in Table 9, it can be seen that the mean precision was 91% and the mean recall was 90%, while the mean F1-score was 89%, which assures the soundness of the system.

#### 4.2.4. Experiment 4: Evaluation Using Other Conventional Systems

The proposed system was also compared with the previously designed state-of-the-art systems with respect to the mean accuracy of activity recognition and localization. The overall activity classification accuracy of the proposed system on the ExtraSensory dataset was 88.25%, while the localization accuracy was 90.63%, comprising a mean accuracy of 89.44%. Regarding the SHL dataset, the activity recognition accuracy was 96.00% and location classification accuracy was 90.50%, making the mean accuracy of the system 93.25%. The mean accuracy of the system was chosen as a comparison parameter for the comparison of the proposed system and the state-of-the-art systems. The comparison statistics that can be examined in Table 10 clearly indicate that our system outperformed the available cutting-edge frameworks.

**Table 2 sensors-23-07363-t002:** Confusion matrix for human activity classification on the ExtraSensory dataset.

	SIT	LYD	STN	BIC	RUN	STL	STU	STD
SIT	0.89	0	0.01	0	0.02	0.02	0.04	0.02
LYD	0.06	0.91	0.03	0	0	0	0	0
STN	0.02	0.02	0.89	0	0.03	0	0.03	0.01
BIC	0	0	0	0.86	0	0	0.09	0.05
RUN	0	0	0	0.04	0.88	0.06	0.02	0
STL	0	0.02	0	0	0.04	0.88	0.03	0.03
STU	0	0	0	0	0.02	0	0.87	0.11
STD	0	0	0	0	0.03	0	0.09	0.88

SIT = sitting; LYD = lying down; STN = standing; BIC = bicycling; RUN = running; STL = strolling; STU = stairs-up; STD = stairs-down.

**Table 3 sensors-23-07363-t003:** Human activity classification report on the ExtraSensory dataset.

Classes	Precision	Recall	F1 Score
Sitting	0.89	0.89	0.88
Lying Down	0.90	0.90	0.88
Standing	0.90	0.89	0.89
Bicycling	0.86	0.86	0.86
Running	0.90	0.89	0.88
Strolling	0.89	0.88	0.88
Stairs-Up	0.89	0.89	0.86
Stairs-Down	0.88	0.88	0.88
Mean	0.89	0.89	0.88

**Table 4 sensors-23-07363-t004:** Confusion matrix for human location classification on the ExtraSensory dataset.

	IND	HOM	SCH	WRK	OUT	CLS	GYM	BCH
IND	0.92	0.02	0	0	0	0	0.04	0.02
HOM	0.03	0.91	0	0.04	0	0	0.02	0
SCH	0.04	0	0.92	0.04	0	0	0	0
WRK	0	0	0.05	0.90	0.03	0.02	0	0
OUT	0	0	0.01	0.04	0.89	0	0	0.06
CLS	0	0	0.04	0.04	0.02	0.90	0	0
GYM	0.05	0.04	0	0	0.02	0	0.89	0
BCH	0	0	0	0.03	0.05	0	0	0.92

IND = indoor; HOM = at home; SCH = at school; WRK = at workplace; OUT = outdoor; CLS = in class; GYM = at gym; BCH = at beach.

**Table 5 sensors-23-07363-t005:** Human location classification report on the ExtraSensory dataset.

Classes	Precision	Recall	F1 Score
Indoor	0.92	0.92	0.91
At Home	0.92	0.91	0.91
At School	0.93	0.91	0.90
At Workplace	0.91	0.90	0.90
Outdoor	0.91	0.89	0.88
In Class	0.90	0.89	0.90
At Gym	0.90	0.88	0.88
At Beach	0.92	0.91	0.90
Mean	0.91	0.90	0.90

**Table 6 sensors-23-07363-t006:** Confusion matrix for human activity classification on SHL dataset.

	SIT	WAL	STN	RUN
SIT	0.95	0.01	0.04	0
WAL	0	0.97	0	0.03
STN	0.06	0	0.94	0
RUN	0	0.02	0	0.98

SIT = sitting; WAL = walking; STN = standing; RUN = running.

**Table 7 sensors-23-07363-t007:** Human activity classification report on SHL dataset.

Classes	Precision	Recall	F1 Score
Sitting	0.96	0.94	0.94
Walking	0.97	0.97	0.96
Standing	0.94	0.94	0.94
Running	0.98	0.98	0.96
Mean	0.96	0.96	0.95

**Table 8 sensors-23-07363-t008:** Confusion matrix for human location classification on SHL dataset.

	IND	OUT	BUS	TRN	SWY	CAR
IND	0.94	0.03	0.01	0	0	0.02
OUT	0.02	0.93	0.05	0	0	0
BUS	0.12	0.01	0.87	0	0	0
TRN	0	0	0.02	0.86	0.11	0.01
SWY	0	0	0	0.13	0.87	0
CAR	0.03	0	0.01	0	0	0.96

IND = indoor; OUT = outdoor; BUS = in bus; TRN = in train; SWY = in subway; CAR = in car.

**Table 9 sensors-23-07363-t009:** Human location classification report on SHL dataset.

Classes	Precision	Recall	F1 Score
Indoor	0.94	0.94	0.94
Outdoor	0.94	0.93	0.91
In Bus	0.88	0.88	0.85
In Train	0.86	0.84	0.84
In Subway	0.87	0.87	0.87
In Car	0.96	0.95	0.93
Mean	0.91	0.90	0.89

**Table 10 sensors-23-07363-t010:** Performance comparison with the state-of-the-art systems.

Methods	ExtraSensory	SHL
Linear Regression [73]	0.83	-
Random Forest [74]	0.87	-
HAR-GCNN [75]	0.87	-
SVM [76]	-	0.79
RNN [77]	-	0.92
ETMD [78]	-	0.92
DNDF (Proposed)	0.89	0.93

## 5. Conclusions

The proposed system deals with human activity recognition and human localization in parallel, while utilizing IoT data from smartphone and smartwatch sensors. The system denoises the data and then splits up into two parallel branches, i.e., the human activity recognition branch and the human localization branch. The respective features are extracted and then passed through a standardization pipeline in which the data are selected and optimized, and the features are sent to two independent DNDFs for their respective classification tasks. The overall performance of the proposed system was better on the SHL dataset as compared to the ExtraSensory dataset, which might be due to the size and complexity of the dataset. The ExtraSensory dataset is larger in size, with 60 subjects participating in the collection of the data, while, for the collection of the SHL dataset, only three subjects contributed. Moreover, the human activities and locations related to the ExtraSensory dataset used in this study were more than those of the SHL dataset, which might also contribute to the higher accuracy of the system on the SHL dataset.

The system aims to achieve a high accuracy by combining multiple windows per stack, extracting reliable features, and utilizing an advanced classification algorithm. Importantly, the system differs from traditional approaches by leveraging IoT data from smartphones and smartwatches. This expands the system’s applicability and provides flexibility in various scenarios.

Although five-second-long windows performed the best for the framework proposed in this study, smaller window sizes also have the potency to prove beneficial for human activity recognition [79]. In the future, we will be looking forward to increasing the system’s robustness by exploring some more effective windowing approaches like applying a dynamic windowing operation on the input data that adapts the window size based on meaningful parameters of the input signal. Moreover, adding some other features in the system for both human activities and human localization will also be under consideration. We also want to add support for more locations and activities in the system and hope to enhance the generalizability of the system by utilizing more advanced deep architectures.

## Figures and Tables

**Figure 1 sensors-23-07363-f001:**
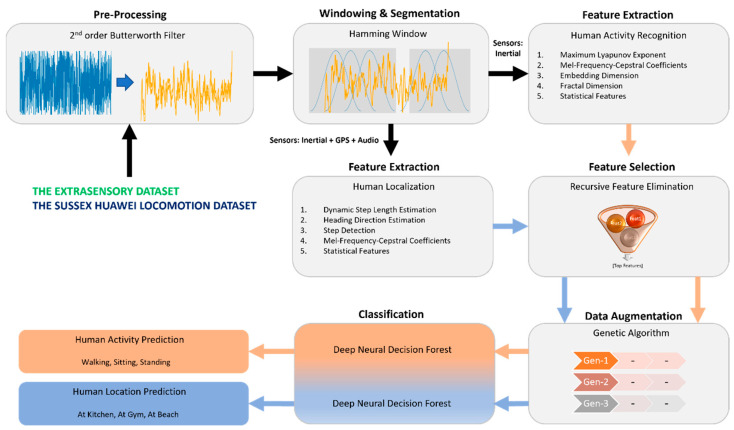
The architecture of the proposed system for human activity recognition and localization.

**Figure 2 sensors-23-07363-f002:**
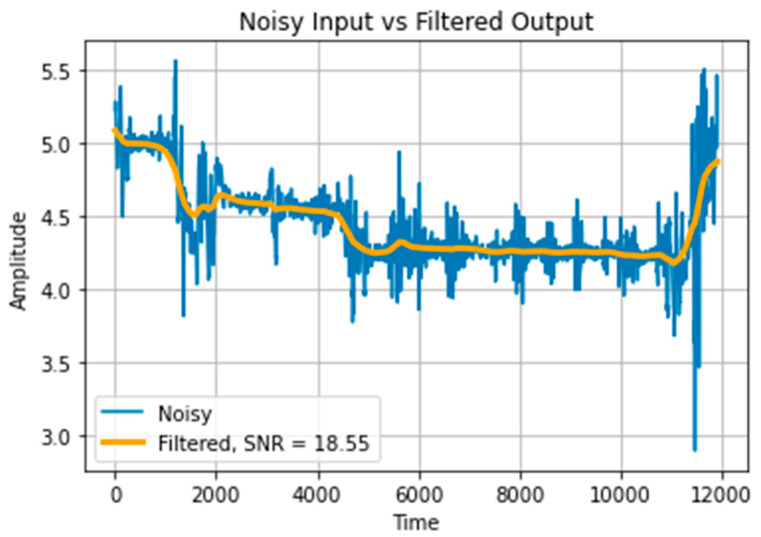
Input signal pre-processing using Butterworth filter.

**Figure 5 sensors-23-07363-f005:**
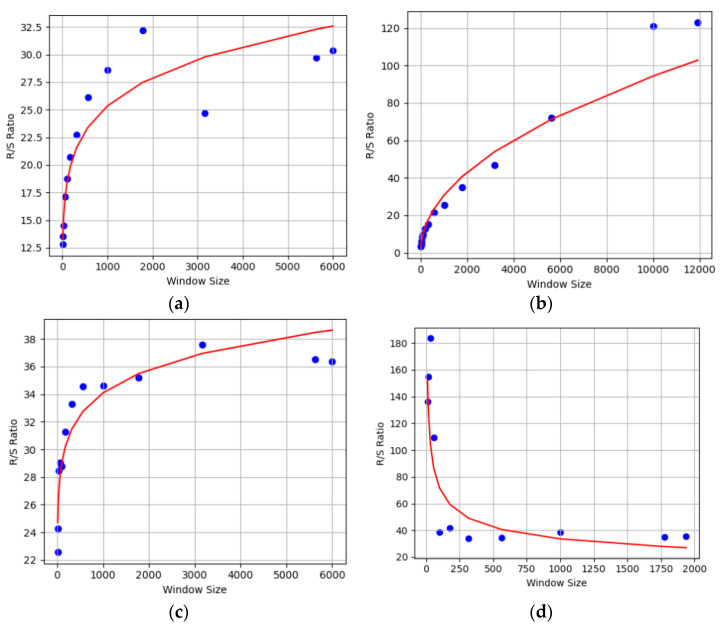
Fractal dimension for (**a**) sitting and (**b**) lying down from ExtraSensory dataset, and (**c**) sitting and (**d**) running from Sussex-Huawei Locomotion dataset.

**Figure 6 sensors-23-07363-f006:**
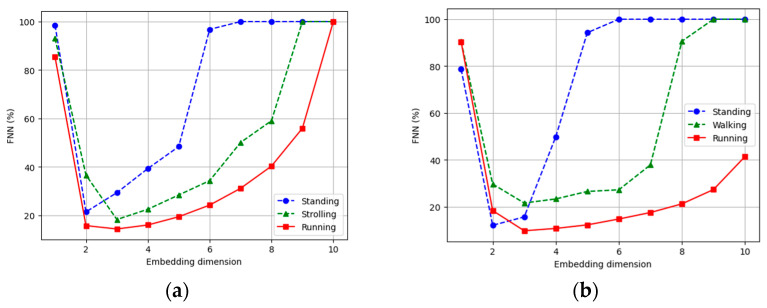
Embedding dimension for (**a**) standing, strolling, and running for the ExtraSensory dataset, and (**b**) standing, walking, and running for the Sussex-Huawei Locomotion dataset.

**Figure 9 sensors-23-07363-f009:**
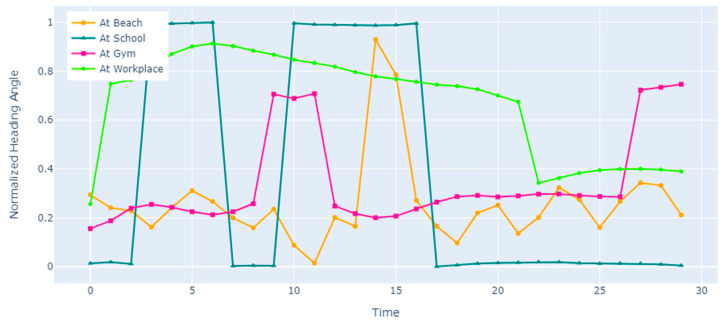
Normalized heading angles for various locations of the ExtraSensory dataset.

**Figure 10 sensors-23-07363-f010:**
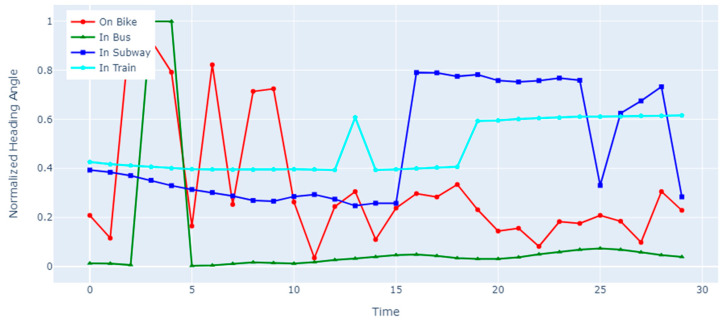
Normalized heading angles for various locations of the Sussex-Huawei Locomotion dataset.

**Figure 13 sensors-23-07363-f013:**
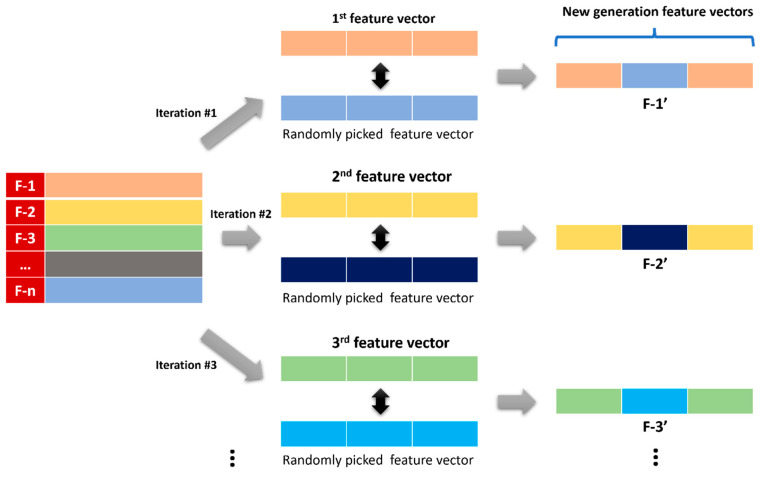
Block diagram for the genetic algorithm as data augmenter.

**Table 1 sensors-23-07363-t001:** Previous works for human activity recognition and localization.

Method	Algorithm	High Points	Limitations	Proposed Solution
Hsu et al. [31]	The sensors measured the accelerations and angular velocities of the human body and transmitted them wirelessly to a computer. The computer then applied a series of steps to process the signals and classified them into different activities using a nonparametric weighted feature extraction algorithm and a principal component analysis method.	Two wearable inertial sensors were used while one was placed on the on the wrist and the other one on the ankle. Sensor data was wirelessly transmitted to the system.	System used only two sensors, which might not capture the full range of human motions and postures. It also requires a wireless connection between the sensors and the computer, which may be unreliable or unavailable in some environments.	More sensors are used to cover different parts of the human body, such as the torso, the backpack, the hand, and the pocket [32]. This allows us to capture more information and the diversity of human motions and postures. Moreover, smartphone embedded sensors have been used so as to recognize the human’s activities and locations, without relying on a wireless connection.
A-Basset et al. [33]	The system is based on heterogeneous human activity recognition (HHAR) and interprets the HHAR as an image classification problem. Their System encodes sensory data into a three-channel (RGB) picture representation and passes it through the system for the activity classification.	System Generated RGB images from HHAR data. Multiscale heirarchical feature extraction. Channel-wise attention unit.	The system was trained on small datasets that makes the generalizability of the system uncertain. Moreover, the computational and space complexity of the system is unclear that makes the scalability of the system uncertain.	Diverse and large datasets were utilized in training of the system that enhances the generalizability of the proposed system. As the system is trained on large datasets, it can handle bigger datasets while maintaining its computational complexity [34].
Konak et al. [35]	The system evaluates the performance of several sets of features taken from accelerometer readings and divides them into three classes: features related to motion, features related to orientation, and features related to rotation. Motion, orientation, and rotational information are used individually and in combination by the system to assess recognition performance. The analysis employs a number of categorization techniques, including decision trees, naive Bayes, and random forests.	Accelerometer based activity recognition. Categorization of the features into rotation, orientation, and motion related features.	Dataset used in the system was collected with the contribution of 10 subjects only that makes the generalizability of the system unceratin. Secondly, They used common machine learning classifiers for the activity reocgnition while advanced models may improve the performance of the system.	The proposed model uses the Extrasensory dataset for training that provides the data of 60 subjects. System achieves state-of-the-art performance over it and proves its ability to be more generalizable. Moreover, system uses a DNDF for the classification that ia an advanced classifier that possess the properties of both machine learning and deep learning classifiers.
Chetty et al. [36]	An innovative data analytic method for intelligent human activity recognition using smartphone inertial sensors was provided. The system used machine learning classifiers such as random forests, ensemble learning, and lazy learning and was based on an information theory-based feature ranking algorithm for the best feature selection.	Smartphone sensors-based activity recognition. Feature ranking based on information theory.	Common machine learning algorithms including lazzy learning, random forest, and ensemble learning were trained on a single dataset. Single dataset might not cover all of the scenarios and can cause system to decay its performance while working in realtime scenarios.	The proposed system is trained on two benchmark datasets that cover a diverse range of activities. Specially, the Extrasensory dataset was collected in wild scenarios when there was not restrictions on the subjects contributing to the data collection. This makes the proposed system more dependable as compared to their system.
Ehatisham-ul-Haq et al. [37]	The framework introduced a novel activity-aware human context recognition method that predicted user contexts based on physical activity recognition (PAR) and learnt human activity patterns in various behavioral circumstances. The method linked fourteen various behavioral situations, including phone positions, with five daily living activities (lying, sitting, standing, walking, and running). Random Forest and other machine learning classifiers were employed in the evaluation of the suggested strategy.	System uses human activity recognition to infer the context of the activity. System also integrates other information like location of the subject and secondary activites like walking while eating, and sitting and talking etc. being performed.	The system mainly depended upon the accelerometer data for the predcition of activities, locations, and secondary activites. While for the location estimation, GPS and microphone data can be a very good addition. Moreover, the system uses a simple random forest for the classification task that can misclassify the complex activites.	The proposed system uses smartphone acclerometer, smartphone magnetometer, smartphone gyroscope, smartwatch accelerometer, smartwatch compas, smartphone GPS, and smartphone’s microphone. By encorporating diverse sensors, the system increases its robustness for the activity recognition and localization. Moreover, DNDF is more advanced than a simple random forest and more reliable in terms of its predictions.
Cao et al. [38]	The system presented an effective Group-based Context-aware classification approach GCHAR, for smartphone human activity recognition. In order to increase classification efficiency and decrease classification errors through context awareness, the system used a hierarchical group-based scheme. GCHAR used context awareness and a two-level hierarchical classification structure (inter-group and inner-group) to identify activity group transitions.	The system proposes a solution to predict human activities along with their contextual information. Inner-group and inter-group classfication approach was adopted to enhance the system’s accuracy.	Their system used tri-axial accelerometer and tri-axial gyroscope to extract the data and process it for the activity classification as well as context awareness. Addition of more sensors can make the performance of the system better.	The proposed system utilizes diverse sensors for the activity recognition and localization. This property makes the proposed system more reliable as compared to their system.
Gao et al. [39]	The research presented a system for jointly recognizing smartphone location and human activity using motion sensors and the multi-task learning (MTL) technique. To combat the detrimental impacts of smartphone orientation change on recognition, the system used a novel data preprocessing technique that included a coordinate modification based on quaternions. The joint recognition model was created to produce results for multiple tasks using a single global model, thereby lowering processing requirements and enhancing recognition efficiency.	Multi-task learning techinuqe. Joint learning of human activities and smartphone location. Data processing based on quaternions.	Their framework used only the motion sensors that could be a drawback especially when classifying the locations of the smartphone.	The proposed system uses GPS and microphone data along with the motion sensors to make the location classification more accurate and reliable.
Fan et al. [40]	In the paper, a Context-Aware Human Activity Recognition (CA-HAR) method was proposed with the goal of identifying human behaviours even while the smartphone was not on the user’s body. The system combined several sensor inputs from the smartphone and used ripple-down rules (RDR) and deep learning to identify activities. In order to solve the on-body location issue, RDR rules were developed using a context-activity model that took into account additional contextual data.	Context-aware human activity recognition based on smartphone sensors. Aggregation of data from multiple smartphone sensors. Ripple down rules to rienforce the correct classification of the activities with context.	Real-time recognition performance may be impacted by the increased computational overhead caused by building and maintaining the context-activity model for RDR rules.	The proposed framework possesses the ability to accurately and robustly predict the human activities and the locations without the need of RDR rules. The extracts such features that generate distinctive representations of the activity examples and then trains a strong classifier such as DNDF for the activity and location classification. All these aspects, make the proposed system work better in challenging scenarios.

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
