# Peer review of "Intelligent Localization and Deep Human Activity Recognition through IoT Devices"

_sensors, 2023, doi:10.3390/s23177363_

Round 1
Reviewer 1 Report
An interesting manuscript.
- Framework -
Figure 1 actually integrates the localization task for a final data augmentation on the recently proposed comprehensive Practical Wearable Sensor-based Human Activity Recognition Research Pipeline, which should be cited.
- Windowing -
First of all, it should only be called windowing here. Segmentation, specifically, is actually another job for HAR.
Secondly, there are great doubts about "five-second intervals and then stack three consecutive windows together". What is the basis for this choice?
1. Recent literature shows that 22 kinds of human daily activities (single motions) have all a duration of 1-2 seconds and are normally distributed in healthy people. 5 seconds actually includes more than one session during a SIT, a LYD, a STN, a BIC, a RUN, a STL, a STU, a STD, etc.
2. Additional considerations: One of the key characteristics and popular application scenarios of smartphone-based HAR/LOCAL is its real-time performance. 5 seconds has actually lost the ability to react immediately, and it is unacceptable to use such an HAR model in interaction or control.
3. If the choice of 5-sec+3-stacking is due to its best (offline) performance among experiments, please evidence it using statistics as a support. In fact, one parameter that you still missed is window overlap. Another thing that can be done is space reduction after stacking (such as using LDA).
I'm not saying that you must supplement more experiments, but just giving 5+3 without explaining the reason makes the method's practicability and reusability greatly reduced. LDA or overlap is something maybe future work, but a parameter selection results of window length + stacking (e.g., using a heatmap) is recommended.
References to cite and follow about joint experiments of window length + overlap + stacking + stacking in HAR:
"Feature Space Reduction for Multimodal Human Activity Recognition"
"Feature Space Reduction for Human Activity Recognition Based on Multi-Channel Biosignals"
In addition, as explained above, it needs to be pointed out in the manuscript that for real-time applications, 5-second windowing is obviously a short come to improve further. How should it be improved in the future? SOTA reference: On a Real Real-Time Wearable Human Activity Recognition System; Interactive and Interpretable Online Human Activity Recognition.
- Features -
There is a essential doubt here. The authors describe the several features they extract, but not why. Are they from literature? Or chosen by yourself? Many HAR studies use time-series feature extraction libraries such as TSFEL, which can easily extract dozens of features in time, frequency, and statistical domain, and a group of features has been proven effective for HAR. In addition, the selection of features, such as greedy forward, ANOVA, MRMR, is also a straightforward task. The author's explanation on features is relatively weak (Biosignal processing and activity modeling for multimodal human activity recognition, Figure 5.22/23).
If the reason for not using enough features is time consumption, "Feature-Based Information Retrieval of Multimodal Biosignals with a Self-Similarity Matrix", Table A1, and "Biosignal processing and activity modeling for multimodal human activity recognition", Table 5.8, provided all features potential for HAR tasks with the lowest computation cost. At least they are references to provide for following experimenters.
- Model -
For DFDF, an image is given but without any detailed description. At least a brief scientific introduction about how you applied the model in your task should be offered.
- Literature -
About performing HAR on IoT Data, SOTAs lately also included
1. HMMs and HHMMs: https://doi.org/10.1007/978-981-19-0390-8_108
2. Pure statistical model without training; automatic activity segmentation and detection for HAR: Feature-Based Information Retrieval of Multimodal Biosignals with a Self-Similarity Matrix
3. Motion Units, up-to-date activity modeling technologies for HAR, e.g., superior accuracy for 17 activities on the smartphone dataset UniMiB (8 fall + 9 daily): Motion Units: Generalized Sequence Modeling of Human Activities for Sensor-Based Activity Recognition.
- English -
There are many grammatical and spelling errors in the manuscript, such as:
Abstract: to tackles
Section 1: THE availability of
THE internet of things
in A pocket or a bag
worth while
Section 2: Smartphone sensors haVE
The above examples are just before the fourth page. I have no interest in listing more, please check carefully.
Importantly, pay attention to the use of English articles. Quite a few "a" or" the" are mistakenly omitted in many places.
See above.
Author Response
Thanks for your valuable comments.
Answers are attached. Please consider.

Reviewer 2 Report
Please Condense the abstract by focusing on the main points and removing introductory text. Also, Include information about data acquisition through IoT in the abstract.
Authors may specify the reason for improving the accuracy of Sussex Huawei Locomotion dataset over the Extrasensory dataset.
Clarify the correct algorithm used for classification and localization, as there is a discrepancy between the mentioned deep neural decision forest algorithm in the introduction and SVM for localization in the fourth bullet text.
Mention the advantages and limitations of using more sensors in the proposed approach within the introduction
Provide the specific reason in section 3.1 for choosing additional data from GPS and microphone for localization.
Author Response

(The authors gave the same response as above.)

Reviewer 3 Report
Dear authors, this is a valuable research work focusing on human activity recognition and localization. The applications of this research include healthcare monitoring, behavior analysis, personal safety, and entertainment. However, there are challenges such as signal noise, smartphone positioning, and accuracy of activity and location predictions. The proposed model employs denoising techniques, segmenting the data, and extracting features for activity recognition and localization to address these challenges. Recursive feature elimination and a genetic algorithm are used for feature selection and data augmentation. The system systematically utilizes a deep neural decision forest to classify activities and locations. The proposed system is evaluated using benchmark datasets, achieving high accuracy rates of 88.25% for activity recognition and 90.63% for location classification on the Extrasensory dataset and 96.00% and 90.50%, respectively, on the Sussex Huawei Locomotion dataset. The present work has archival value and should be considered for publication after the authors address a few comments I have.
1. The paper contains numerous typographical errors. The authors should meticulously review the paper to rectify all of them.
2. In a research paper, using the term 'our' is excessively informal and should be avoided.
3. The excessive use of the pronoun 'we' is observed throughout the paper. It is best to use 'we' when discussing future work in the conclusion. However, apart from that, it should be used sparingly.
4. Certain sentences tend to be excessively lengthy. Constructing concise sentences, each conveying a single idea, is generally preferable.
5. Certain visual representations appear blurry. The authors should either employ higher-resolution figures or recreate them as vector graphics.
6. The authors should enhance the explanation of the research's context, including why the research problem holds significance.
7. The introduction should explain the principal limitations of previous works relevant to this paper.
8. The authors should explain the distinctions between prior works and the solution presented in this paper.
9. To highlight the dissimilarities and limitations, the authors should incorporate a table comparing previous works' key characteristics. Additionally, they may consider adding a row in the table to describe the proposed solution.
10. Although a groundbreaking solution is put forth, it is imperative to offer better explanations for the design choices, such as the rationale behind the solution's specific design.
11. It is vital to explicitly clarify what aspects of the proposed solution are novel and what aspects are not. In cases where certain portions are identical, appropriate citations should be made, and the differences should be emphasized.
12. It is necessary to delve into the complexity of the proposed solution and discuss it in detail. Moreover, the discussion should also involve comparisons with other solutions available in the literature.
13. Need more discussion on the experimental results.
14. The following additional experiments are necessary:
- Runtime
- Memory
15. To ensure the reproducibility of the results, the authors should publicly release the code of the proposed solution on a website.
1. The paper contains numerous typographical errors. The authors should meticulously review the paper to rectify all of them.
2. In a research paper, using the term 'our' is excessively informal and should be avoided.
3. The excessive use of the pronoun 'we' is observed throughout the paper. It is best to use 'we' when discussing future work in the conclusion. However, apart from that, it should be used sparingly.
4. Certain sentences tend to be excessively lengthy. Constructing concise sentences, each conveying a single idea, is generally preferable.
Author Response

(The authors gave the same response as above.)

Round 2
Reviewer 1 Report
The authors addressed the raised questions well and made a significant improvement.
Though the window length selection has been rationalized by greedy selection on 1-6 seconds, one thing remains to be clarified, as mentioned in Review round 1, that is the potency of shortening the 5-second window to endow the model with meaningful real-world duration on activity modeling and real-time capability. Just as review round 1 said, "Recent literature shows that 22 kinds of human daily activities (single motions) have all a duration of 1-2 seconds and are normally distributed in healthy people. 5 seconds actually includes more than one session during a SIT, a LYD, a STN, a BIC, a RUN, a STL, a STU, a STD, etc." Ref./Citation: How Long Are Various Types of Daily Activities?
Moreover, Sec. 2.1 and 2.2 seem to be only two tables. They should be captioned. It is recommended to merge the two sections into one and write a sentence to refer to the two tables. A table, without any format of a table, as the main textual part of an article, is not acceptable for publication. Not to mention that the whole subsection is just a table with nothing else to introduce it. Incidentally, mentioning Sec. 2.1, https://doi.org/10.5220/0007398800470055 (BIOSTEC 2019 best student paper) is an important IoT HAR work using knee bandage as a novel IoT sensors carrier, extending IoT sensor placement introduction (smartphone, torso, backpack, hand, pocket...) in Sec. 2.1.
To save the time and academic resources of the authors and the editorial offices, I do not insist on opening another round of reviewing. I argue to accept this manuscript after addressing the above minors.
Author Response
Thanks for your valuable comments.
All comments are addressed in attached file. Please consider.
